# Silicon: quantum dot photovoltage triodes

Wen Zhou [1], Li Zheng [1✉], Zhijun Ning [2✉], Xinhong Cheng[1], Fang Wang[3], Kaimin Xu[2], Rui Xu[2], Zhongyu Liu[2], Man Luo[3], Weida Hu [3], Huijun Guo[3], Wenjia Zhou [2] & Yuehui Yu[1]

Silicon is widespread in modern electronics, but its electronic bandgap prevents the detection of infrared radiation at wavelengths above 1,100 nanometers, which limits its applications in multiple fields such as night vision, health monitoring and space navigation systems. It is therefore of interest to integrate silicon with infrared-sensitive materials to broaden its detection wavelength. Here we demonstrate a photovoltage triode that can use silicon as the emitter but is also sensitive to infrared spectra owing to the heterointegrated quantum dot light absorber. The photovoltage generated at the quantum dot base region, attracting holes from silicon, leads to high responsivity (exceeding 410 A·W$^{-1}$ with $V_{bias}$ of $-1.5$ V), and a widely self-tunable spectral response. Our device has the maximal specific detectivity (4.73 × 10$^{13}$ Jones with $V_{bias}$ of $-0.4$ V) at 1,550 nm among the infrared sensitized silicon detectors, which opens a new path towards infrared and visible imaging in one chip with silicon technology compatibility.

[1] State Key Laboratory of Functional Materials for Informatics, Shanghai Institute of Microsystem and Information Technology, Chinese Academy of Sciences, Shanghai 200050, P. R. China. [2] School of Physical Science and Technology, ShanghaiTech University, Shanghai 201210, P. R. China. [3] State Key Laboratory of Infrared Physics, Shanghai Institute of Technical Physics, Chinese Academy of Sciences, Shanghai 200083, P. R. China. ✉email: zhengli@mail.sim.ac.cn; ningzhj@shanghaitech.edu.cn

Microelectronics, the miniaturization of silicon integrated circuits, make it possible to low-cost and high-performance digital imaging[1–3]. Despite the plentiful virtues of silicon, its 1.1 eV bandgap limits the infrared sensing range and communication applications[4–6]. Heterogeneous integration of non-silicon infrared sensitive materials with silicon is an effective and promising way to expand the wavelength range from the visible to short-wave infrared range (SWIR)[7]. Germanium and III–V compounds have been employed in the heteroepitaxy on silicon but the fabrication complexity increases due to high-temperature epitaxy and silicon contamination mitigation processes[8,9]. The OD materials, such as the colloidal quantum dot (CQD), are the promising candidates for the next-generation light-sensing materials due to their low-cost manufacturing, facile solution processability, tunable bandgap, and flexible substrate compatibility[10,11]. Besides, they can be easily integrated with silicon-based read out circuits (ROIC) by printing or spin-coating[12]. The CQD photodiodes (PDs), photoconductors (PCs), and photovoltage field-effect transistors (PVFETs) have been reported to show the versatility of a CQD film and its flexibility to other semiconductors[4–6,13–20]. However, PDs, whether based on CQD homojunctions or CQD: other semiconductor heterojunctions, produce no gain. As a result, the spectral responsivity is extremely low[5,6,18]. PCs or hybrid transistors can significantly improve the gain owing to the differential time between the capture time of one photon-generated carrier and the transit time across the channel of the other one[11,19]. However, the specific spectral detectivity is generally low due to the normally-on channel induced serious dark current[17–20]. PVFETs based on the photovoltage generated at the CQD:silicon interface, combined with the high transconductance provided by the epitaxial silicon device, have been proposed to achieve high gain and a widely tunable spectral response (from visible light to 1500 nm)[4]. This front exploration shows that CQDs can be used as an efficient platform for silicon-based infrared detection. Nonetheless, the dark current density of the prototype PVFET reaches $10^{-1}$–$10^{1}$ A cm$^{-2}$, which is still too high for a practical silicon-based photoelectric chip[3].

The current working mechanism of the Si:CQD infrared detectors is using silicon for charge transport and CQDs for the light absorber[4–6,14,21], which means the photo-generated charges should overcome the interfacial energy barrier to transfer from CQDs to silicon. The energy barrier from CQDs to silicon will increase with the decreased bandgap of CQDs[5,22,23], but unfortunately, the absorption cutoff wavelength is also inversely proportional to the bandgap of CQDs. As a result, the above working mechanism will hamper Si:CQD detectors to further extend their detection in the infrared range.

Here we have demonstrated a Si:CQD photovoltage triode (PVTRI). Photo-induced carriers can generate at the n-CQD:p-CQD junction interface and are separated by the built-in potential. The photoelectrons can unimpededly transfer into the base region (n-CQD) but will be blocked off by the conduction band offset at the p-Si:n-CQD heterojunction. As a result, a negative photovoltage arises at the base region to control the Si:CQD junction electrostatics. Holes in silicon can be attracted by the photovoltage and transfer to CQDs. The Si:CQD PVTRI shows both high responsivity and specific detectivity in the infrared (1550 nm) and visible (637 nm) light in excess of $10^{2}$ A W$^{-1}$ and $10^{13}$ Jones, respectively. The dark current densities are $10^{-9}$–$10^{-3}$ A cm$^{-2}$ for a bias voltage of $V_{bias} = 0$ V to $-1.5$ V, which are at least four orders of magnitude lower than that of previous infrared sensitized silicon detectors[4]. We explain and demonstrate the physical principles that govern the operation of the PVTRI using simulations, band models and experiments, and show the potential of the device for infrared and visible imaging in one chip with state-of-the-art silicon techniques.

## Results

**Structure of the PVTRI.** Our Si:CQD PVTRI arrays have been fabricated on a 4-inch silicon wafer shown in Fig. 1a. The minimum pixel size is $40 \times 40$ μm (Fig. 1b) and the schematic device structure of a single Si:CQD PVTRI is illustrated in Fig. 1c. A lightly doped p-Si substrate contacted with ohmic aluminum acts as the emitter at $V_{bias} < 0$. The base and collector are consisted of ultrathin PbS n-CQD (0.16 μm) and p-CQD (0.04 μm) films with iodide (PbS-I) and 1,2-ethanedithiol (EDT) ligands, respectively, at this case. Indium tin oxide (ITO) has a high transmission coefficient in the visible and SWIR wavelengths (Fig. S1), which serves as a transparent electrode in the PVTRI. The substrate terminal is connected to the ground plate while the bias is applied to the ITO top electrode, and the vertical current is monitored under illumination and in the dark. Figure 1d reveals the thicknesses and highlights the uniform character of each layer in the Si:CQD PVTRI by the cross-sectional scanning electron micrograph (SEM). The composition is also quantified by energy dispersive X-ray spectroscopy (EDS) (Fig. S2). Figure 1e, f shows the distribution morphology and size of PbS CQDs on the Si substrate by transmission electron microscopy (TEM) and high-resolution transmission electron microscopy (HRTEM), respectively. The crystal structure of the PbS CQDs on Si is confirmed via selected area electron diffraction (SAED) as shown in Fig. 1g and the SAED pattern demonstrates the high crystallinity of the PbS CQDs. In addition, the positions of the X-ray diffraction (XRD) peaks (Fig. 1h) agree well with the standard diffraction patterns of the face-centered cubic PbS[24,25].

To investigate the doping type and elementary composition of the PbS CQDs, the PbS-I and PbS-EDT CQDs films have been analyzed via X-ray photoelectron spectroscopy (XPS). Figure 1i shows the elementary composition of PbS-TBAI and PbS-EDT CQDs. As shown in Fig. 1g, S $2p_{1/2}$ and $2p_{3/2}$ peaks with respective binding energies of 163.2 eV and 164.5 eV corresponding to S-C bonds only appear in PbS-EDT. This is in consistent with as reported p-type doping of PbS CQDs[26,27]. As for PbS-TBAI, the I 3d region has well separated spin–orbit components by 11.5 eV (one peak at 619 eV and the other at 630.5 eV), which demonstrates stable n-type doping of PbS CQDs[25]. Figure 1k shows the absorption spectra of silicon and PbS CQDs from visible to SWIR wavelengths. Silicon has excellent visible light absorption but does not have detectable absorption at the wavelength beyond 1100 nm. PbS CQDs have a broad spectra absorption range from the visible light to 1600 nm range with an exciton peak of 1500 nm.

**Simulation of the PVTRI.** Technology computer aided design (TCAD) Sentaurus simulations have been performed to illustrate the working principles of the device. A low bias of $V_{bias} = -0.01$ V has been applied on the PVTRI. Upon 1550 nm incident illumination (Fig. 2a), photo-induced carriers are generated exclusively within the n-CQD:p-CQD junction ($\sim 10^{21}$ cm$^{-3}$s$^{-1}$). Only a small amount of photoelectrons can cross the energy barrier between n-CQD and silicon and increase the electron density in silicon. The majority of photoelectrons aggregate at the base region and at the interface of Si:n-CQD (Fig. 2b). Photoelectrons produce a negative photovoltage at the interface via the photovoltaic effect (Fig. 2c). In the PVTRI, this effective bias shrinks the depletion region in the silicon, attracts holes transferring from the silicon emitter to the base region (Fig. 2d) and produces an extra current detected by the collector.

**Physical principles and PVTRI operation.** We also analyze the operation of the PVTRI to further explain the physical mechanisms that govern its behavior by the band offset theory. The band

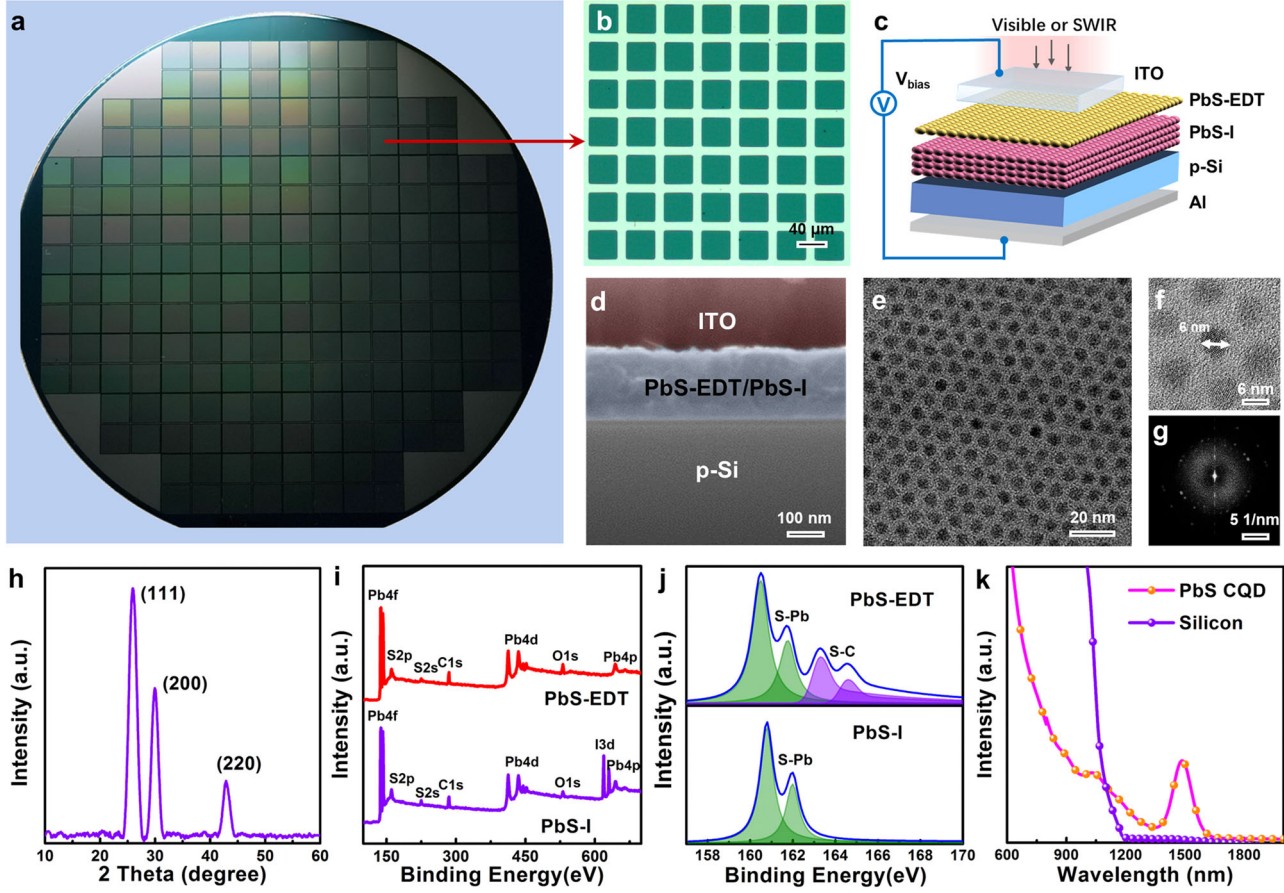

**Fig. 1 Structure and material characterization of the PVTRI. a** PVTRI arrays in a 4-inch silicon wafer. **b** The optical microscope image to show a minimum pixel (40 × 40 μm) of the arrays. **c** The schematic device structure of a single Si:CQD PVTRI. At $V_{bias} < 0$, the p-Si substrate acts as the emitter while the base and collector are consisted of the ultrathin PbS n-CQD and p-CQD films, respectively. **d** The cross-sectional SEM image of the PVTRI. TEM (**e**) and HRTEM (**f**) images with the SAED patterns (**g**) of the PbS CQDs. **h** XRD analysis of PbS CQDs. XPS elementary composition (**i**) and S 2p peak spectra (**j**) of PbS-I and PbS-EDT. **k** Absorption spectra of the PbS CQDs and silicon.

gap of PbS CQDs is 0.78 eV, extracted from the absorption spectra (Fig. 1i). The valence ($\Delta E_{vo}$) and conduction ($\Delta E_{co}$) band offsets between n-PbS and p-silicon are 0.16 eV and 0.18 eV, respectively, (details of the energy band barriers calculation of the Si:n-PbS heterojunction seen in Supplementary Note 1). Both of the p-Si:n-CQD heterojunction and the n-CQD:p-CQD junction show the rectifying characteristics (Fig. S3).

The photocurrent amplification of the PVTRI can be realized by applying a negative bias at the collector with the emitter grounded ($V_{bias} < 0$ V). The p-Si:n-CQD heterojunction is forward biased while the n-CQD:p-CQD junction is reverse biased (Fig. 3a). Holes from p-Si can unimpededly transport to n-CQD and be extracted to p-CQD by the reversed p-CQD:n-CQD junction. The voltage drop on the two p–n junctions can be estimated from the I–V curves (Fig. S3). The dynamic resistances of the junctions are calculated by $R = dV/dI$. As shown in Fig. S3a, for the n-CQD:p-Si heterojunction, the dynamic resistance is in the range of 0.06–1.28 kΩ cm² when the voltage changes from 0 V to −1 V. However, for the p-CQD:n-CQD junction (Fig. S3b), its dynamic resistance is in the range of 10.83–95.56 kΩ cm² when the voltage changes from 0 V to −1.5 V, which is one to two orders of magnitude higher than the one of the n-CQD:p-Si heterojunction. At 1.5 V ≤ $V_{bias}$ < 0 V, the resistance of the heterojunction is at least ten times than the resistance of the homojunction, and the voltage across the heterojunction is at most 0.13 V. Such a low voltage cannot guarantee the n-CQD:p-Si heterojunction turn on and no bias

voltage is applied on the base region (n-CQD) to amplify $I_{CE}$. As a result, the p-Si:n-CQD heterojunction is not completely turned on and the dark current at this case is weak (Fig. 3c). Details of the theoretical calculation of the dark current are in Supplementary Note 2.

Under SWIR illumination, the photo-induced electron–hole pairs are mainly separated at the reversed p-CQD:n-CQD junction. The photoholes can be extracted by the reversed junction electric field and collected by the electrode. The photoelectrons are blocked off at the p-Si and n-CQD heterojunction, accumulate, and produce a negative photovoltage at the interface via the photovoltaic effect (the same effect that produces an open-circuit voltage in solar cells)[20,28–30]. The photovoltage controls the Si:CQD heterojunction electrostatics, shrinks the depletion region of p-Si, and continuously extracts the holes from the emitter until the illumination is shut down. As a result, a high gain will be achieved due to the hole current induced by the photovoltage combined with the current produced by the photoholes.

We also discuss the PVTRI with a positive bias ($V_{bias} > 0$ V) to systematically and comprehensively analyze its operation mechanism. At this case, p-CQD is the emitter, n-CQD is the base and p-Si is the collector. The p-Si:n-CQD heterojunction is reverse biased while the n-CQD:p-CQD junction is forward biased (Fig. 3b) at this case. The dark current is also weak because the forward n-CQD:p-CQD junction is not completely turned on along with the valence band offset induced barrier at the reversed

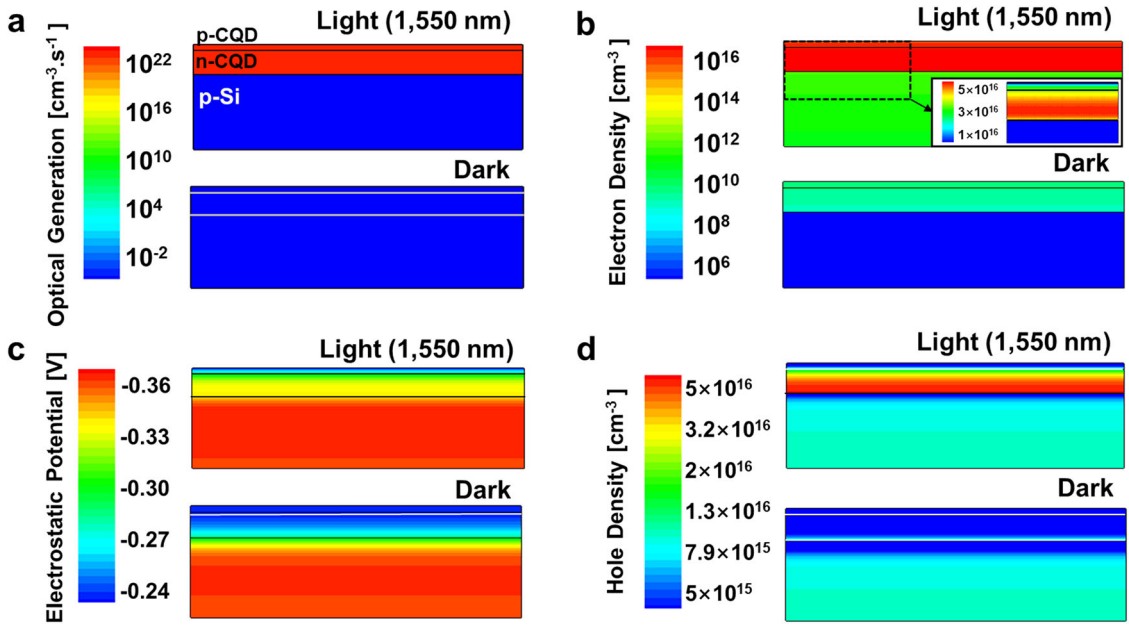

**Fig. 2 Sentaurus TCAD simulations of the PVTRI.** The device parameters are shown in Table S1. **a** Optical generation under illumination of 1550 nm compared with the equilibrium in dark. Photo-included carriers are generated in the p-CQD and n-CQD layers. **b** Electron density of the PVTRI. The majority of photoelectrons aggregate at the base region and at the interface of Si:n-CQD. **c** Electrostatic potential of the PVTRI. A higher negative electrostatic potential (photovoltage) is obtained under illumination of 1550 nm. **d** Hole density of the PVTRI. The negative photovoltage at the interface can attract holes transferring from silicon to the base region under the illumination of 1550 nm.

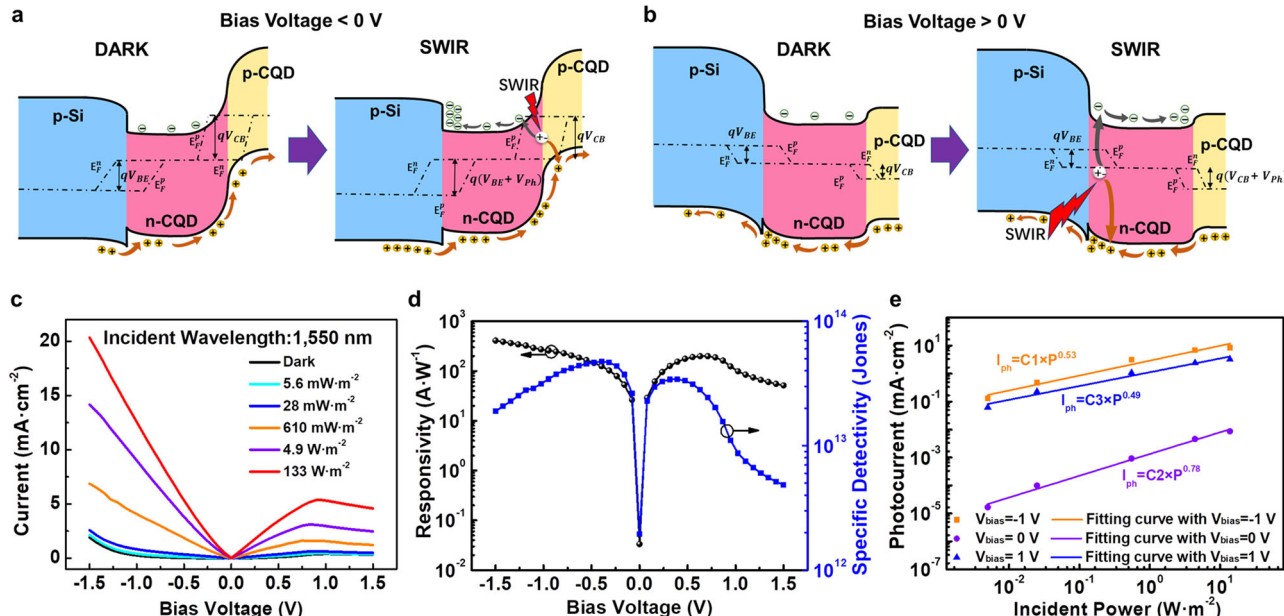

**Fig. 3 Operation of the PVTRI with excitation of SWIR.** Energy bands and operation mechanism of the PVTRI with **a** negative bias ($V_{bias}$ < 0 V) and **b** positive bias ($V_{bias}$ > 0 V) voltages with excitation of SWIR and in the dark, respectively. **c** Output characteristics of the PVTRI under 1550 nm illumination with different incident powers. **d** The responsivity and specific detectivity of the PVTRI as a function of bias voltages at low incident power of 5.6 mW·m$^{-2}$. **e** The photocurrents as functions of incident powers at 0, −1, and 1 V, respectively.

p-Si:n-CQD heterojunction. Under SWIR illumination, photo-induced carriers can only generate at the n-CQD side of the reversed heterojunction. The photoelectrons will be blocked off at the n-CQD:p-CQD junction and a negative photovoltage is produced at the interface, shrinks the depletion region of n-CQD:p-CQD junction, and extracts holes from p-CQD to the base region (n-CQD). However, only part of the holes can transport across the valence band offset induced barrier by

tunneling and the other holes are blocked off at the n-CQD:p-Si interface. As a result, the photocurrent and gain will be less than the case of $V_{bias}$ < 0 V.

The operational behavior of the PVTRI is firstly investigated at 1550 nm excitonic wavelength illumination. Figure 3c shows the I–V curves of the PVTRI with different bias voltages and incident powers. The experiments results indicate that the photocurrent at a negative bias ($V_{bias}$ < 0 V) is notably higher than the one at a

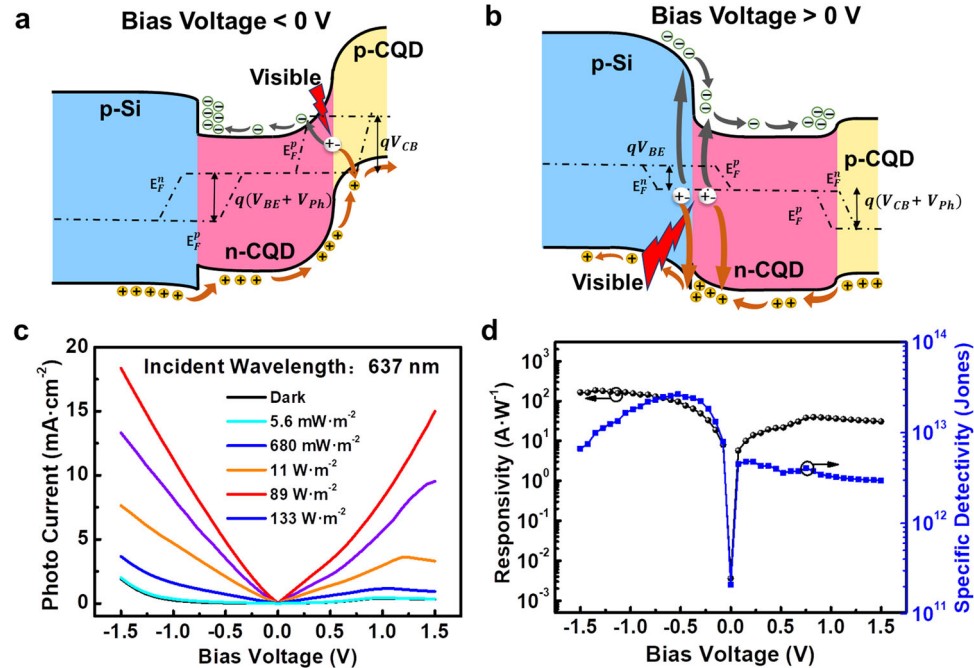

**Fig. 4 Operation of the PVTRI with visible excitation.** Energy bands and operation mechanism of the PVTRI with **a** negative bias and **b** positive bias voltages in the visible excitation. **c** Output characteristics of the PVTRI with different incident powers under illumination of 637 nm. **d** The responsivity and specific detectivity of the PVTRI as a function of bias voltages at a low incident power of 5.6 mW m$^{-2}$.

positive bias ($V_{bias} > 0$ V). This is exactly consistent with the theoretical analysis discussed in the above section. For the negative bias, the photocurrent rises with increasing bias. The maximum responsivity is 410 A W$^{-1}$ at a bias of −1.5 V, giving rise to a high gain of 330 and specific detectivity of $1.92 \times 10^{13}$ Jones (Fig. 3d). The gain is calculated by the equation $G = (I_{ph}/q)/(P/h\nu) = R \times h\nu/q$, where $h\nu$ is the incident photon energy, $q$ is the unit charge, $I_{ph}$ is the photocurrent, $P$ is the incident power, and $R$ is the responsivity ($R = I_{ph}/P$)[31]. In addition, the specific detectivity can be further increased by bias tuning and reach up to $4.73 \times 10^{13}$ Jones at a bias of −0.4 V (with the responsivity of 125 A W$^{-1}$). The specific detectivity of our PVTRI is an order of magnitude higher than the previous reported highest value of the Si:CQD photodetectors[4].

It is worthy to mention that for the positive bias, the photocurrent first rises, reaches to the saturation, and then gradually falls with increasing bias. The non-monotonic behavior of the device current detected at $V_{bias} > 0$ V is due to the large injection effect. The device current density is directly proportional to the small-signal common-base gain ($\beta$). At the case of low injection, $\beta$ will increase with $V_{bias}$ increasing. When $V_{bias}$ exceeds a threshold, the injected minority carrier concentration will rapidly increase, which can be equal to or even larger than the majority carrier concentration and lead to more sharp increase of $J_{nE}$ than $J_{pE}$, where $J_{pE}$ and $J_{nE}$ are the current densities due to the diffusion of the minority carriers in the base and emitter, respectively (details seen in Supplementary Note 2). According to Eq. S6, $\beta$ will decrease at the case of large injection with $V_{bias}$ increasing. This is one reason why the device current shows a non-monotonic behavior for $V_{bias} > 0$ V. The second reason is that at the case of high injection, the minority carrier in the space charge region of the base cannot be completely depleted, which increases the neutral region width in the base and moves the depletion region edge in the base towards the collector[32]. This effect will also reduce $\beta$, and lead to the non-monotonic behavior of the device current. In addition, increasing $V_{bias}$ can enhance the electronic field of the Si:CQD heterojunction, and the

photoinduced electrons are easier to be extracted. The photo-induced electrons are blocked off at the interface of the n-CQD:p-CQD junction to increase photovoltage initially. However, with $V_{bias}$ increasing, the barrier of the n-CQD:p-CQD junction will be gradually lowered down. More photoelectrons can tunnel across the barrier of the n-CQD:p-CQD junction and reduce the photovoltage, which also results in $\beta$ decrease. As a result, it is a trade-off between the photocurrent and the positive bias voltage. In addition, the saturation knee point of the I–V curve (right branch of Fig. 3b) positively shifts with the illumination power increasing. It is because higher illumination power induces higher photovoltage, which needs more positive bias voltage to weaken it.

Figure 3e shows the relationship between the incident power and the photocurrent at zero ($V_{bias} = 0$ V), positive ($V_{bias} = 1$ V) and negative ($V_{bias} = -1$ V) bias voltages. The relation curves can be fitted by the equation $I_{ph} = c \times P^k$, where $c$ and $k$ are the empirical values[33]. The defects induced capturing, trapping, and releasing of the charge carriers will make the $k$ value less than 1[34,35]. The $k$ values obtained from the fitting curves are 0.53 and 0.49 at $V_{bias} = 1$ V and −1 V, respectively, while the $k$ value (0.78) for the case at $V_{bias} = 0$ V is significantly greater. It seems that applying extra electronic states will complicate the photoelectric conversion process and extend the trapping time in CQD and it is verified by the temporal response of the PVTRI in the next section.

We then studied the performance of the PVTRI under visible and near-infrared illumination. As shown in Fig. 4a, at a negative bias, the visible photo-induced electron–hole pairs are mainly separated at the reversed p-CQD:n-CQD junction. Similar to the case with SWIR illumination, a negative photovoltage generates at the n-CQD:p-CQD junction to amplify the photocurrent. At a positive bias (Fig. 4b), photo-induced electron–hole pairs will generate at both sides of the reversed heterojunction and the forward homojunction, respectively. However, the homojunction is forward biased and the space charge region width ($W$) narrows with $V_{bias}$ increasing. The photo-induced electron–hole pairs will

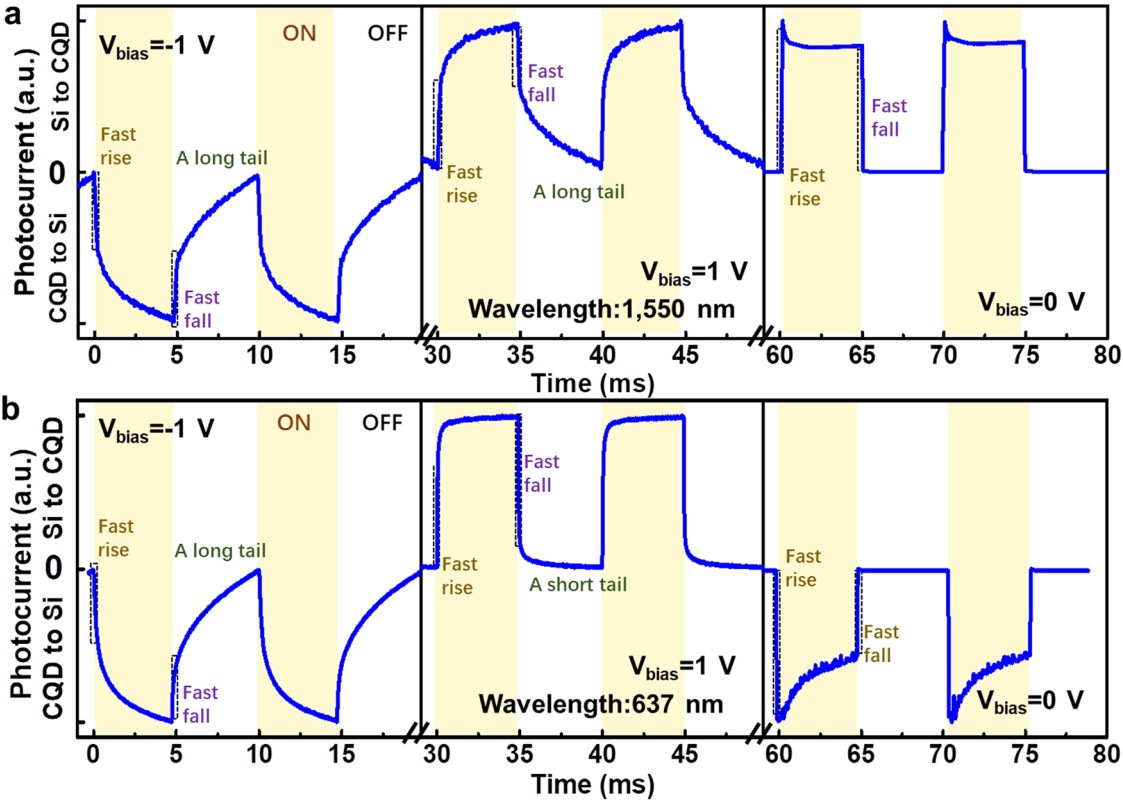

**Fig. 5 Response time of the PVTRI.** Response time of the PVTRI under illumination of **a** 1550 nm and **b** 637 nm, respectively.

recombine soon at this case, and the photocurrent generated at the forward homojunction is negligible compared with the one at the reversed heterojunction. As shown in Fig. 4c, the visible light (637 nm) excited photocurrent is much larger than the wavelength of 1550 nm at the positive bias, which attributes to the higher visible optical absorption rate of the p-Si:n-CQD heterojunction. At a low bias of $-1.35$ V, the responsivity (Fig. 4d) are up to 186 A W$^{-1}$, with a high gain of 362 and specific detectivity of $1.02 \times 10^{13}$ Jones. The specific detectivity can reach up to $2.47 \times 10^{13}$ Jones at a bias of $-0.5$ V (with the responsivity of 101 A W$^{-1}$). In the case of near-infrared excitation (1060 nm and 1310 nm in Fig. S5), our PVTRI can also obtain high responsivity and specific detectivity on the order of $10^2$ A W$^{-1}$ and $10^{13}$ Jones, respectively.

**Temporal performance and self-tunable spectral response.** Figure 5a shows the temporal photocurrent response curves of the PVTRI under illumination of 1550 nm with applied negative ($V_{bias} = -1$ V), positive ($V_{bias} = 1$ V), and zero bias ($V_{bias} = 0$ V) voltages at room temperature. At a negative bias voltage, similar to the PVFET[4], the PVTRI also has fast-response and slow-response components. The fast-response component is related to the device structure and the slow-response component (a long tail) is related to the defects of the materials (e.g., defect states in the CQD films and interface)[4,36,37]. As for the fast-response component, it shows both fast rise (50 μs) and fall (50 μs) edges (Fig. S6). This component of temporal response is usable for general sensing and imaging devices[3,33], which has a much faster response time than the traditional CQD-based photoconductors[34] and phototransistor[17,19] (about 100 ms). The defects-related long tail can be reduced by further improving the synthesis method of CQDs.

At the case of applying a positive bias voltage, a similar fast rise (100 μs) and fall (70 μs) edges with a long tail are detected, but the direction of photocurrent (from silicon to CQD) is opposite to the one with a negative bias voltage (from CQD to silicon). It has been verified that external electric field can introduce complicated electronic states in the CQD photodetector[20,38]. Removing these electronic states could shorten the response time of our PVTRI towards 9 μs rise and 6 μs fall with no detectable tail at zero bias (Fig. S6c), making it closer to the photodiode and enabling applications such as time-of-flight sensing and machine vision. The response time can be further improved by promoting the CQD quality and reduce its trapping state in the synthetic processing[4]. Noise performance of the detector has been measured as a function of frequency (Fig. S7). The specific detectivity can also be calculated by the normalization noise current according to the following equations: $D^* = R / \sqrt{I_n^{*2}}$ and $I_n^* = I_n / \sqrt{A}$, where $I_n^*$ and $I_n$ are the normalization noise current and the noise current, respectively; $R$ is the responsivity and $A$ is the device area[39–41]. $I_n$ generally consists of the shot noise current ($I_{ns}$), the generation–recombination (G–R) noise current ($I_{ngr}$) and the 1/f noise (flicker noise) current ($I_{nf}$) according to the following equations: $I_n = \sqrt{(I_{ns})^2 + (I_{ngr})^2 + (I_{nf})^2}$, so $D^*$ may be over-estimated if only accounting for the shot noise. In this work, $D^*$ is calculated by using the normalization noise current ($I_n^*$) at the cut-off frequency ($f_T$). Details of the calculation approach are justified in Supplementary Note 3. As shown in Fig. S7, the calculated total normalization noise current curve is basic anastomotic with the measured one. In addition, the calculated total normalization noise current at $f_T$ is $1.01 \times 10^{-11}$ AHz$^{-0.5}$cm$^{-1}$, which is very close to the measured one ($1.17 \times 10^{-11}$ AHz$^{-0.5}$cm$^{-1}$). According to the measured $I_n^*$ at $f_T$, the extracted specific detectivity of the

PVTRI at 1550 nm excitonic wavelength is up to $1.07 \times 10^{13}$ Jones at $V_{bias} = -0.4$ V, which is of the same magnitude as the value calculated by the dark current method.

We investigated the temporal response of the PVTRI at 637 nm excitation to show its distinguishable response time under SWIR and visible light illumination. At the case of applying a negative bias (Fig. 5b), the temporal response is very similar to the one at 1550 nm excitation because the reversed n-CQD:p-CQD junction dominates the photoelectric conversion. However, if the applied bias is positive, significantly faster rise (8 μs) and fall (5 μs) with a short tail are detected (Fig. S6e), revealing the dominant role of the reversed p-Si:n-CQD heterojunction in the photoelectric conversion. The commercial silicon with much lower defects and higher visible light absorption makes the photo-induced carriers less likely to be trapped by the detect states. As for the case with no bias voltage, similar fast response has been detected with an opposite direction of the photocurrent. The distinguishable response time between SWIR and visible light excitation endows our PVTRI the self-tunable capability to distinguish SWIR and visible signals in one chip, which is promising in the cutting-edge optoelectronic applications. If SWIR and visible light do not reach the device at the same time, they can be distinguished by the falling edge tail of the $I_{ph}$-$t$ plot at the case of $V_{bias} > 0$ V. There is distinguishable response time under SWIR and visible light illumination for the proposed device. At $V_{bias} > 0$ V, the response time (falling edge) of visible light is significantly faster than the one of SWIR. If SWIR and visible light reach the device at the same time, they can be distinguished by double measurements on both sides of the device. To begin with, the position of the device can be adjusted to make the incident light illuminate on the Si substrate. At this case, the Al electrode should not cover the whole substrate for a real-life application. Since the Si substrate is hundreds of microns, it can act as a visible light filter and we can get the signal of SWIR by this measurement. Then, we can reverse the device and make the incident light illuminate on the CQD. The mixed signal of SWIR and visible light can be got by this second measurement. At last, we can get the signal of visible light by curve extraction from the above two plots.

The opposite direction of photocurrent under illumination of 1550 nm and 637 nm with zero bias ($V_{bias} = 0$ V) voltages can be explained by the energy band theory (Fig. S8). Under SWIR illumination, photo-induced carriers generate at the n-CQD:p-CQD junction and are separated by the built-in potential. Photoholes can be extracted to the ITO electrode, but most photoelectrons are blocked off by the barrier of the Si:n-CQD heterojunction. As a result, the photocurrent at this case is low with a direction from Si to CQD. If the PVTRI is illuminated under visible lights, photo-induced carriers will generate at both p-Si:n-CQD and n-CQD:p-CQD junctions with opposite photocurrent directions. The direction of the total photocurrent ($I_{ph} = I_{ph,EB} - I_{ph,CB}$) is opposite to the case under SWIR illumination due to higher visible light absorption of silicon than PbS CQDs.

Figure 6 illustrates the specific detectivity and responsivity of the representative reported PbS CQDs and infrared sensitized silicon photodetectors[4–6,14,21,42–46]. Compared with the previous reported photodetectors, the sensitization of our PVTRI does not rely on any interface engineering or high-temperature epitaxial growth, has a high specific detectivity ($4.73 \times 10^{13}$ Jones at $V_{bias}$ of $-0.4$ V) at 1550 nm, and opens a new path towards infrared and visible imaging in one chip with silicon technology compatibility.

## Discussion

In summary, we have designed and fabricated a Si:CQD infrared photodetector with a ultrahigh specific detectivity of $4.73 \times 10^{13}$

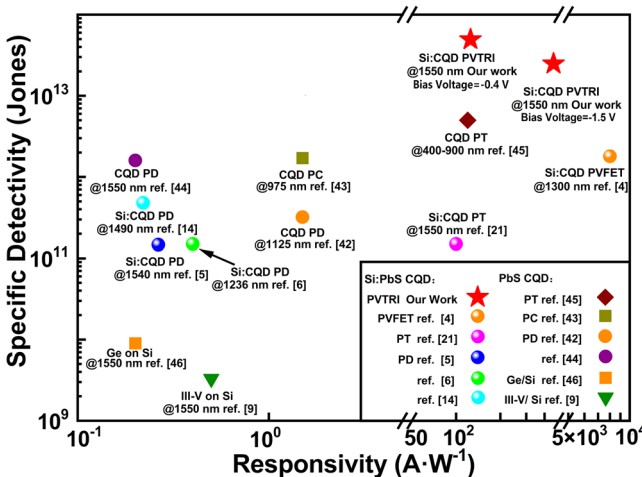

**Fig. 6 Performance comparision of the representative PbS CQDs and silicon photodetectors.** Responsivity and detectivity comparison between our PVTRI and the representative reported PbS CQDs and infrared sensitized silicon photodetectors.

Jones (at $V_{bias}$ of $-0.4$ V) at 1550 nm for the first time. This high detectivity is enabled by a high responsivity at the order of magnitude of $10^2$ A W$^{-1}$ associated with a low dark current. Comprehensive analysis of the operation mechanism indicates the construction of a photovoltage triode structure. This structure addresses the trade-off of the low responsivity of the photodiode, and the large dark current of the photoconductor or phototransistors. More importantly, this device structure is made on a silicon chip that is fully compatible with the commercial CMOS ROIC. Furthermore, the PVTRI shows bias controllable response time at fixed wavelengths and distinguishable response time between SWIR and visible light excitation. The PVTRI structure brings a new freedom for performance enhancement of CQD-based infrared photodetectors.

## Methods

**Synthesis of PbS CQDs.** Lead oxide (0.900 g, 4 mmol), oleic acid (OA, 28 g) and octadecene (ODE, 20 g) were mixed together to a two necked flask and heated under vacuum at 110 °C for 3 h. Afterwards, these mixtures were kept at 120 °C under N$_2$ flow to form a clear solution. 420 mL (2.02 mmol) of bis(trimethylsilyl) sulfide was dissolved in 20 mL of ODE and injected into the two necked flack. After 30 s, the heating jacket was quickly removed and the mixed solution was naturally cooled down. When it reached room temperature, with the addition of 30 mL of acetone and centrifuged for 5 min at 4200 rpm, the nanocrystals were isolated away from the solution. After that, the nanocrystals were purified by dispersion in octane (6 mL) and reprecipitated with acetone (40 mL per time) three times to remove OA and ODE solution. Finally, the nanocrystals were dried under vacuum at room temperature for 20 min and dissolved in octane at a concentration of 50 mg per mL.

**N-type and p-type CQDs films fabrication.** We explored n-type and p-type doping of PbS CQDs by TBAI and EDT ligands exchange, respectively. The n-type and p-type PbS CQD films were fabricated by a layer-by-layer method. For n-type CQDs, PbS CQDs solution was spun on the substrate for 25 s at 2500 rpm. After that, the TBAI (10 mg mL$^{-1}$) solution was dropped to exchange the ligand for 30 s before being spun cast, followed by two rinse spin steps using methanol. The thickness of each layer is about 20 nm. For p-type CQDs, the PbS CQDs solution was also spun on a substrate for 25 s at 2500 rpm. After that, the EDT (0.02 vol%) solution was dropped to exchange the ligand for 15 s before being spun cast, followed by two rinse spin steps using acetonitrile. The thickness of each layer is also about 20 nm.

**PVTRI fabrication.** The lightly doped p-type Si (100) substrate was cleaned with the 4:1 H$_2$SO$_4$:H$_2$O$_2$ solution for 10 min to remove the organic contamination. Then the substrate was dipped in the BOE: H$_2$O (4:100) solution to remove the native oxide and rinsed with deionized water. Afterwards, 8 layers (~160 nm) of n-type PbS CQDs and 2 layers (~40 nm) of p-type PbS CQDs were spun layer-by-layer on the silicon substrate, followed by annealing at 80 °C for 10 min to

evaporate the organic solvent and increase the stability of CQDs. An aluminum electrode was deposited on the backside of silicon by the thermal evaporation and the ITO electrode was deposited on the PbS-EDT layer surface by the magnetron sputtering. The ITO and PbS CQDs were patterned by lithography and etched by the ion beam etching for isolation among the PVTRI arrays.

**Photodetector characterization**. The photocurrent was measured by the optical electric analyzer platform (Agilent B1500 and Thorlabs ITC4001). From the range of visible to SWIR region, four typical lasers (Thorlabs LPS series) with wavelengths of 637, 1060, 1310, and 1550 nm were used. The transient responses of the photodetectors were also measured by the optical electric analyzer platform and dual-channel picoammeter voltage source (Keithley 6482), and the on and off times of the laser were controlled by a mechanical shutter.

## Data availability
All data supporting the findings of this study are available within the article and/or the supplementary information. The raw data is available from the corresponding author upon reasonable request.

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

## Acknowledgements
This work was supported by the National Natural Science Foundation of China (Grant No. 12075307), Shanghai Rising-Star Program (21QA1410900), the Science and Technology Commission of Shanghai Municipality (Grant Nos. 19511107400, 20501110900 and 20501110800), Chinese Academy of Sciences Programs (Grant Nos. KFJ-STS-QYZD-132, YJKYYQ20190025 and 172231KYSB20190004), and Shanghai Sailing Program (Grant No. 20YF1456700). We would like to thank Prof. Xiaoming Xie, Prof. Zengfeng Di, Prof. Wenjie Yu, Dr. Wanning Xu, Mr. Xiongbin Xiao, Mr. Liang Ke, and Mr. Tengfei Xu for their generous help.

## Author contributions
L.Z., X.C., and Y.Y. directed the research work. L.Z. and W.Z. conceived and designed the experiments and simulation. K.X., R.X., and W.-J.Z. conducted the synthesis of the colloidal materials. W.Z. and Z.L. fabricated the devices and performed the TEM, SEM, XPS, XRD, and transmission spectrum characterization. F.W. and H.G. performed the noise current measurement. W.Z., M.L., W.H., and Z.N. performed the photoelectric measurements. L.Z. and W.Z. analyzed the data and designed the figures. W.Z., L.Z., Z.N., and W.H. co-wrote the manuscript.

## Competing interests
The authors declare no competing interests.
