## [Peer Review File · Nature Communications]

Reviewers' Comments:

Reviewer #1:

Remarks to the Author:

In this work, the authors provide a comprehensive investigation over a triode photodetector based on Si: quantum dot heterojunction. The working mechanism for the photodetector can be controlled using different voltage biases. The photodetector also has a high detectivity and micro-second response time. The work is convincing and meaningful, and I think it will be of interest for the researchers in the field. Here are some minor issues to be addressed before acceptance.

- 1) In abstract section, the highest responsivity and detectivity are obtained under different voltage bias conditions (-1.5V and -0.4V), which must be clarified.
- 2) In line 207, equation for detectivity should be given.
- 3) In lines 203 and 207, what are the detectivity at -1.5V and responsivity at -0.4V?
- 4) In line 244, what are the detectivity at -1.35V and responsivity at -0.5V?
- 5) In figure 6, are the highest responsivity and detectivity obtained at different bias conditions? If so, please only plot responsivity and detectivity values at same voltage bias.
- 6) in line 315, the authors claimed the device requires no interface engineering. However, the authors' previous work (<https://pubs.acs.org/doi/10.1021/acsami.0c01744>) claims that without appropriate Si surface treatment (CHI3), charge transport between Si and quantum dots will be impeded, resulting in negligent EQE at SWIR region, which seems to be contradictory. Can authors address this?
- 7) Can authors provide a spectral EQE plot?
- 8) In line 259 and figure 5, how did authors define rise & fall times. Rise time normally is defined as the time for signal to rise from 10% to 90% of the signal value. Based on this, in figure 5, it seems that the response time including the long tail is in ms order, much longer than 50 us claimed in the work.

Reviewer #2:

Remarks to the Author:

The paper "Silicon:Quantum Dot Photovoltage Triodes" reports about a phototransistor built upon the heterojunction between Silicon and PbS Quantum Dots. The proposed device structure is quite novel, nevertheless other photodetectors exploiting the Si/QD heterostructures have been described in the past years.

Even though such a photodetector can be of some interest for the academic community involved in the study of QD-based devices, the paper is not accurate enough and appears speculative especially in the part dedicated to the device's working principle. For these reasons I believe the paper should be rejected. Here I attach a detailed list of the most important issues affecting the paper.

1. Even though the authors refer to the proposed device as a "triode", it seems to me it should be viewed as a classic heterojunction bipolar transistor (HBT), where base and collector are made of PbS QDs, while the emitter is realized in the silicon substrate. The word "triode" is misleading since it is typically employed to describe vacuum-tube amplifiers.
2. The explanation of the device's working principle is unclear. In particular:
 - 2.1 For $V_{ce} < 0$ the sentence "the p-Si:nCQD heterojunction is not completely turned on and the dark current at this case is weak" is too speculative and unclear. The authors neglect the different electrostatic characteristics of the materials composing the device. Since the authors employed TCAD simulations in the first part of the manuscript, they could have used the same approach to

estimate the voltage drop on the two pn junctions at a given V_{ce} bias. More importantly, the photocurrent gain observed in the case of $V_{ce} < 0$ could be explained by the theory of operation of HBTs. In this case, a simple calculation, accounting for the difference between Si and CQD bandgaps, gives an expected photocurrent gain between 200 and 500 depending on the diffusion length of the electrons in the Si substrate (for this calculation, I used the data reported by the authors in table 1). This result is completely compatible with the measurements reported in the paper.

2.2 For $V_{ce} > 0$ the sentence "the photoelectrons will be blocked off at the n-CQD:p-CQD junction and a negative photovoltage is produced at the interface, shrinks the depletion region of the n-CQD:p-CQD junction and extract holes from p-CQD to the base region (n-CQD)" is not clear. According to the band diagram of fig. 3b, it is not clear why only the photoelectrons should be blocked at the n-CQD:p-CQD interface. Moreover, the authors assume that holes are transferred from the n-CQD to the p-Si by tunneling, but this assumption is not substantiated by experimental data nor theoretical analysis (i.e. give the low barrier height at the Si:CQD heterojunction also thermal emission phenomena should be considered). Finally, it is not clear why, at $V_{ce} > 0$, the authors still observe a photocurrent gain > 1 .

2.3 The device current shows a non-monotonic behavior for $V_{ce} > 0$ both in dark and under illumination. The authors discuss this behavior, but their explanation seems speculative. No numerical, experimental, or theoretical analysis are provided to substantiate the authors' assumptions. Also, references are lacking.

2.4 Fig. 4b and the relative discussion are misleading, since it seems that visible photons are absorbed only at the p-Si:n-CQD interface. The sentence "At a positive bias, photo-induced electron-hole pairs will generate at both sides of the reversed heterojunction" is also misleading, since it neglects carrier generation occurring in the forward bias homojunction.

3. The authors claim to have reached very high detectivity values (4.73×10^{13} Jones in the SWIR and 2.47×10^{13} Jones in the visible), but it is unclear how they evaluated the noise current. First, it seems the authors assume shot noise is the only type of noise affecting the device performance while, in the "Temporal performance and self-tunable spectral response" paragraph, the authors discuss the need to also consider other types of noise in their calculation. In this second case it is not clear how the authors could obtain a detectivity as high as 1.33×10^{13} Jones. In fact, when dealing with flicker noise, it is important to consider the integral of the noise current over the whole bandwidth of the device. Looking at the normalization noise current reported in the supplementary information it is clear that the device is strongly affected by flicker noise, but the authors apparently consider only the noise current measured at a single frequency corresponding to the cut-off frequency of the device. In this case, the authors are assuming to couple the device to an ideal bandpass filter, centered at the cut-off frequency. This assumption seems unrealistic and should have been explained in more detail in the manuscript. Moreover, the reported noise data has been measured at very low V_{ce} ($-0.1V$) while the responsivity is always discussed at higher voltages. Since the flicker noise current may be nonlinearly dependent on the DC current flowing in the device, the noise should be evaluated at the same bias at which the responsivity is measured. My opinion is that the calculated detectivity is way too optimistic and a thorough noise analysis should be conducted before claiming such high detectivity values.

4. The authors claim that "The distinguishable response time between SWIR and visible light excitation endows our PVTRI the self-tunable capability to distinguish SWIR and visible signals in one chip, which is promising in the cutting-edge optoelectronic applications". This assumption should be discussed in more details, since it is not clear how the authors would employ the proposed device in a real-life application in order to distinguish between different wavelengths. What happens if SWIR and visible light reach the device at the same time?

5. Some references are out of context and should be removed.

6. The paper is poorly written and should be thoroughly checked for language inconsistencies.

Response to reviewers:

We would like to thank the reviewers for their highly constructive reviews of our manuscript. In response to these comments and suggestions, we have carried out extensive additional experiments, acquired new data and conducted further analysis, and thus we have substantially enhanced our results and understanding. We have made a thorough revision of our manuscript accordingly and addressed all the concerns made by the reviewers. In the following, we address each of the reviewers' comments in detail.

Point-to-Point Responses

Reviewer #1 (Remarks to the Author):

In this work, the authors provide a comprehensive investigation over a triode photodetector based on Si: quantum dot heterojunction. The working mechanism for the photodetector can be controlled using different voltage biases. The photodetector also has a high detectivity and micro-second response time. The work is convincing and meaningful, and I think it will be of interest for the researchers in the field. Here are some minor issues to be addressed before acceptance.

Response: We do thank the reviewer for his/her very positive comments on the significance of our work.

1. *In abstract section, the highest responsivity and detectivity are obtained under different voltage bias conditions (-1.5 V and -0.4 V), which must be clarified.*

Response: We agree with the reviewer that the highest responsivity and detectivity, which are obtained under different voltage bias conditions (-1.5V and -0.4V), must be clarified in abstract section. We have clarified the description in the abstract section.

2. *In line 207, equation for detectivity should be given.*

Response: We agree with the reviewer that in line 207, equation for detectivity should be given. We have added the equation for detectivity ($R = I_{ph}/P$) in the revised manuscript.

3. *In lines 203 and 207, what are the detectivity at -1.5 V and responsivity at -0.4 V?*

Response: The maximum responsivity is $410 \text{ A}\cdot\text{W}^{-1}$ at a bias of -1.5 V, giving rise to a high gain of 330 and specific detectivity of 1.92×10^{13} Jones. In addition, the specific detectivity can be further increased by bias tuning and reach up to 4.73×10^{13} Jones at a bias of -0.4 V (with the responsivity of $125 \text{ A}\cdot\text{W}^{-1}$). We have added the values of the specific detectivity at -1.5 V and responsivity at -0.4 V in the revised manuscript.

4. *In line 244, what are the detectivity at -1.35 V and responsivity at -0.5 V?*

Response: At a low bias of -1.35 V, the responsivity are up to $186 \text{ A}\cdot\text{W}^{-1}$, with a high gain of 362 and specific detectivity of 1.02×10^{13} Jones. The specific detectivity can reach up to 2.47×10^{13} Jones at a bias of -0.5 V (with the responsivity of $101 \text{ A}\cdot\text{W}^{-1}$). We have added the values of the specific detectivity at -1.35 V and responsivity at -0.5 V in the revised manuscript.

5. In Figure 6, are the highest responsivity and detectivity obtained at different bias conditions?
If so, please only plot responsivity and detectivity values at same voltage bias.

Response: We agree with the reviewer's point, and we have modified Figure 6 to show the responsivity and detectivity values at same voltage bias.

Fig. R-1 Responsivity and detectivity comparison between our PVTRI and the representative reported PbS CQDs and infrared sensitized silicon photodetectors.

6. In line 315, the authors claimed the device requires no interface engineering. However, the authors' previous work (<https://pubs.acs.org/doi/10.1021/acsami.0c01744>) claims that without appropriate Si surface treatment (CHI_3), charge transport between Si and quantum dots will be impeded, resulting in negligent EQE at SWIR region, which seems to be contradictory. Can authors address this?

Response: We thank the reviewer for this comment. The physical mechanism of our PVTRI is totally different from the heterojunction-based photodiode in our previous work (<https://pubs.acs.org/doi/10.1021/acsami.0c01744>). **The valence (or conduction) band offset induced barrier between p-Si:n-CQD (or n-Si:p-CQD) will block holes (or electrons) transferring from CQD to Si, but will not block holes (or electrons) transferring from Si to CQD.**

For the p-Si:n-CQD (or n-Si:p-CQD) photodiode, photon-induced carriers are generated in the CQD under 1550 nm illumination. Taking the p-Si:n-CQD photodiode for example, as shown in Fig. R-2(a), photoholes in CQD should overcome the valence band offset induced barrier at the interface to **transfer from CQD to Si**. As a result, appropriate Si surface treatment should be performed to lower the barrier and enhance EQE.

As for the p-Si:n-CQD:p-CQD PVTRI, its working principle exactly takes advantage of the barrier at the interface. When applying a negative bias at p-CQD with p-Si grounded ($V_{bias} < 0$ V), the p-Si:n-CQD heterojunction is forward biased while the n-CQD:p-CQD junction is reverse biased. As shown in Fig. R-2(b), **holes from p-Si can unimpededly transport to n-CQD** and be extracted to p-CQD by the reversed p-CQD:n-CQD junction. The photoelectrons are blocked off at the p-Si and n-CQD heterojunction interface due to the valence band offset induced barrier. The accumulated photoelectrons can produce a negative photovoltage at the interface via the

photovoltaic effect (the same effect that produces an open-circuit voltage in solar cells), which is similar to applying a negative voltage to the base (n-CQD). The photovoltage controls the Si:CQD heterojunction electrostatics, shrinks the depletion region of p-Si, and continuously **extracts the holes from Si to CQD** under 1550 nm illumination. As a result, the claim that the device requires no interface engineering is not in contradiction with our previous work.

Fig. R-2 Energy bands of (a) p-Si:n-CQD photodiode and (b) p-Si:n-CQD:p-CQD PVTRI with a negative bias voltage.

7. Can authors provide a spectral EQE plot?

Response: We thank the reviewer for this suggestion. The spectral EQE plot is as follows and we have added this plot in the Supplementary Information (Supplementary Figure 9).

Fig. R-3. Spectral EQE plot of the PVTRI.

8. In line 259 and Figure 5, how did authors define rise & fall time. Rise time normally is defined as the time for signal to rise from 10% to 90% of the signal value. Based on this, in Figure 5, it seems that the response time including the long tail is in ms order, much longer than 50 us claimed in the work.

Response: We thank the reviewer for this comment. There are two ways to define the response time for a conventional photodetector. One is that the response time is defined as the time for signal to rise from 10% to 90% (or fall from 90% to 10%) of the signal value. The other is that the response time is defined as the time for the photocurrent to rise to $(1-e^{-1})=63\%$ (or fall to $e^{-1}=37\%$) of the maximal photocurrent. **In this manuscript, we have referred to Sargent's method [Ref. R1] to distinguish the fast-response component and slow-response component (a long**

tail) of the response time. The fast-response and slow-response components indicate two different physical mechanisms. The $I-t$ curve can be fitted with a biexponential relaxation equation as follows

$$I = I_0 + A \exp(-t/\tau_1) + B \exp(-t/\tau_2) \quad (\text{R1})$$

where I_0 is the steady-state photocurrent, t is the time, A and B are the constants, and τ_1 and τ_2 are the relaxation time constants corresponding to fast-response and slow-response components, respectively [Ref. R2, R3]. **The fast-response component is related to the device structure and the slow-response component is related to the defects of the materials (e. g. defect states in the CQD films and interface).** The fast-response of rise and fall edges (Fig. 5 of the manuscript) indicates that our proposed device itself can achieve rapid photoresponse, and this is the key point we would like to state. As for the defects-related long tail, it can be reduced by improving the synthesis method of CQDs.

We have added the above explanation and discussion in the manuscript to make it better reasoned.

Ref. R1 Adinolfi, V. and Sargent, E. H. Photovoltage field-effect transistors. Nature 542, 324-327 (2017).

Ref. R2. L. Xuan, et al. Ultrahigh-responsivity, rapid-recovery, solar-blind photodetector based on highly nonstoichiometric amorphous gallium oxide. ACS Photonics, 4, 2203-2221 (2017).

Ref. R3 S. Oh, et al. Development of solar-blind photodetectors based on Si-implanted β -Ga₂O₃. Opt. Express, 23, 28300-28305 (2015).

Reviewer #2 (Remarks to the Author):

The paper “Silicon:Quantum Dot Photovoltage Triodes” reports about a phototransistor built upon the heterojunction between Silicon and PbS Quantum Dots. The proposed device structure is quite novel, nevertheless other photodetectors exploiting the Si/QD heterostructures have been described in the past years.

Even though such a photodetector can be of some interest for the academic community involved in the study of QD-based devices, the paper is not accurate enough and appears speculative especially in the part dedicated to the device’s working principle. For these reasons I believe the paper should be rejected. Here I attach a detailed list of the most important issues affecting the paper.

Response: We do thank the reviewer for his/her recognition of the novelty of the proposed device structure in our work. Although Si/QD heterostructures have been described in the past years, the physical mechanism of our PVTRI is totally different from the reported Si/QD heterostructure-based photodetectors, and we have addressed this question in the response to the Reviewer #1’s Comment 6. In addition, we are very sorry for some of the unclear statements which make the reviewer misunderstand several key points of our work. We have carefully studied the reviewer’s comments and have made correction which we hope meet with approval. On behalf of my co-authors, we solemnly state that the paper is definitely not speculative.

1. Even though the authors refer to the proposed device as a “triode”, it seems to me it should be viewed as a classic heterojunction bipolar transistor (HBT), where base and collector are made of PbS QDs, while the emitter is realized in the silicon substrate. The word “triode” is misleading since it is typically employed to describe vacuum-tube amplifiers.

Response: For the silicon-based devices, “triode” is not just to describe vacuum-tube amplifiers. “Phototriode” has been used to describe the photodetector which is configured in the common emitter mode with the base open circuited since 1960s [Ref. R4, R5]. In addition, “Phototriode” has also been used to describe other semiconductor (MAPbBr₃) n-p-n photodetector [Ref. R6]. There are two reasons why we use “photovoltage triode” rather than HBT to name the proposed device.

- i. As for HBT, the emitter and the base materials form the heterojunction and in order to lower the injection deficit, the band gap of the emitter material should be wider than the base one [Ref. R7, R8]. In our proposed p-Si:n-CQD:p-CQD PVTRI, the bias voltage is applied on p-CQD with p-Si grounded. If using p-Si as the emitter, n-CQD as the base and p-CQD as the collector, the device structure is similar to a HBT (left branch of Fig. SR3). However, if using p-CQD as the emitter, n-CQD as the base and p-Si as the collector, as shown in the right branch of Fig. SR3, the device still has very high responsivity and specific detectivity (e.g. $R=156 \text{ A}\cdot\text{W}^{-1}$ and $D^*=3.43\times 10^{13}$ Jones at V_{bias} of 0.4 V). At this case, the emitter (p-CQD) and the base (n-CQD) are both PbS and have the same bandgap. Therefore, HBT is not appropriate to name the proposed device.
- ii. The gain mechanism of the proposed device is similar to a phototriode, using the base-collector diode as a photodiode and amplifying the photocurrent of this diode by base potential regulation. It is worthy to mention that the barrier at the interface of p-Si and n-CQD will block the photoinduced electrons transferring from n-CQD to p-Si but will not hamper the hole transferring from p-Si to n-CQD. When applying a negative bias, the blocked photoelectrons at the p-Si and n-CQD heterojunction interface will accumulate and

produce a negative photovoltage at the interface via the photovoltaic effect (the same effect that produces an open-circuit voltage in solar cells), which is similar to applying an extra negative voltage to the base (n-CQD). Therefore, we do think “photovoltage triode” is appropriate to name the proposed device.

Fig. R-3. The responsivity and specific detectivity of the PVTRI as a function of bias voltages at low incident power of $5.6 \text{ mW} \cdot \text{m}^{-2}$.

Ref. R4 V. I. Turkulets, N. P. Udalov, *Photodiodes and phototriodes*, Gosénergoizdat, Moscow-Leningrad (1962).

Ref. R5 N. D. Potekhina, *Calculation of relaxation processes of a phototriode for low-intensity illumination*, Soviet Physics-Solid State, Amer. Inst. Physics (1960).

Ref. R6 F. -X. Liang, *Fabrication of MAPbBr₃ single crystal p-n photodiode and n-p-n phototriode for sensitive light detection application*, Adv. Funct. Mater., 30, 2001033 (2020).

Ref. R7 H. Kroemer, *Theory of a wide-gap emitter for transistors*, Proceedings of the IRE, 45, 1535-1537 (1957).

Ref. R8 T. Sugii, et al. *Polycrystalline SiC for a wide-bandgap emitter of Si-HBTs*, J. Electrochem. Soc., 136,3111-3115 (1989).

2. The explanation of the device's working principle is unclear. In particular:

2.1 For $V_{ce} < 0$ the sentence “the p-Si:n-CQD heterojunction is not completely turned on and the dark current at this case is weak” is too speculative and unclear. The authors neglect the different electrostatic characteristics of the materials composing the device. Since the authors employed TCAD simulations in the first part of the manuscript, they could have used the same approach to estimate the voltage drop on the two pn junctions at a given V_{ce} bias. More importantly, the photocurrent gain observed in the case of $V_{ce} < 0$ could be explained by the theory of operation of HBTs. In this case, a simple calculation, accounting for the difference between Si and CQD bandgaps, gives an expected photocurrent gain between 200 and 500 depending on the diffusion length of the electrons in the Si substrate (for this calculation, I used the data reported by the authors in table 1). This result is completely compatible with the measurements reported in the paper.

Response: We are sorry for the speculative and unclear statements and we do thank the reviewer

for his/her affirmation of the compatibility between his/her calculation results with our measurements reported in the manuscript. We have modified the statements and added more theoretical explanation in the manuscript and supplementary information.

i. Estimation of the voltage drop on the two p-n junctions at a given V_{bias}

The voltage drop on the two p-n junctions can be estimated from the I-V curves (Fig. R-4). The dynamic resistances of the junctions can be calculated by $R=dV/dI$. As shown in Fig. R-4a, for the n-CQD:p-Si heterojunction, the dynamic resistance is in the range of 0.06-1.28 $k\Omega\text{cm}^2$ when the voltage changes from 0 V to -1 V. However, for the p-CQD:n-CQD junction (Fig. R-4b), its dynamic resistance is in the range of 10.83-95.56 $k\Omega\text{cm}^2$ when the voltage changes from 0 V to -1.5 V, which is one to two orders of magnitude higher than the one of the n-CQD:p-Si heterojunction. Therefore, at $V_{bias} < 0$ V, the resistance of the heterojunction is at least ten times than the resistance of the homojunction, and the voltage across the heterojunction is at most 0.13 V. Such a low voltage cannot guarantee the n-CQD:p-Si heterojunction turn on with no illumination, so the dark current at this case is weak. The detailed calculation of the dark current will be discussed in the next section.

Fig. R-4. The junction characteristics of the p-Si:n-CQD heterojunction and the n-CQD:p-CQD junction.

ii. Theoretical calculation of the dark current and gain at $V_{bias} < 0$ V

As for a triode, the small-signal common-base gain β is defined as

$$\beta = \frac{J_{pE} - J_{rB}}{J_{nE} + J_{rB}} \quad (R2)$$

Ignoring the base region recombination current, eq.R-2 can be derived as

$$\beta \approx \frac{J_{pE}}{J_{nE}} \quad (R3)$$

where J_{pE} is the current density due to the diffusion of the minority carriers (holes) in the base, J_{nE} is the current density due to the diffusion of the minority carriers (electrons) in the emitter.

As for a p-n-p triode, J_{pE} , J_{nE} and the reverse-biased saturation current density in the base-collector junction (J_S) can be expressed by the following equations [Ref. R9]

$$J_{pE} = \frac{eD_B n_{B0}}{L_B} \left(\frac{1}{\sinh \frac{x_B}{L_B}} + \frac{e^{V_{BE}/k_B T} - 1}{\tanh \frac{x_B}{L_B}} \right) \approx \frac{eD_B n_{Bi}^2}{L_{eff-B} N_B} e^{V_{BE}/k_B T} - 1 \quad (R4)$$

$$J_{nE} = \frac{eD_E n_{E0}}{L_E} \frac{e^{V_{BE}/k_B T} - 1}{\tanh \frac{x_E}{L_E}} \approx \frac{eD_E n_{Ei}^2}{L_{eff-E} N_E} e^{V_{BE}/k_B T} - 1 \quad (R5)$$

$$J_s = \frac{eD_B p_{B0}}{L_B} + \frac{eD_C n_{C0}}{L_C} = \frac{eD_B n_{Bi}^2}{L_B N_B} + \frac{eD_C n_{Ci}^2}{L_C N_C} \quad (R6)$$

where $D_{E/B/C}$ is the minority carrier diffusion coefficients in the emitter/base/collector, $n_{Ei}/p_{Bi}/n_{Ci}$ is the intrinsic carrier concentrations in the emitter/base/collector, $N_{E/B/C}$ is the doping concentrations in the emitter/base/collector, and $L_{\text{eff-E/B/C}}$ is effective minority carrier diffusion lengths in the emitter/base/collector.

According to the above equations, β can be written as

$$\begin{aligned} \beta &= \frac{n_{Bi}^2 N_E D_B L_{\text{eff-E}}}{n_{Ei}^2 N_B D_E L_{\text{eff-B}}} = \frac{(m_{nB}^* m_{pB}^*)^{\frac{3}{2}} N_E D_B L_{\text{eff-E}} e^{\frac{\Delta E_g}{k_B T}}}{(m_{nE}^* m_{pE}^*)^{\frac{3}{2}} N_B D_E L_{\text{eff-B}}} \\ &\approx \frac{N_E D_B L_{\text{eff-E}} e^{\frac{\Delta E_g}{k_B T}}}{N_B D_E L_{\text{eff-B}}} \end{aligned} \quad (R7)$$

Similarly, the dark current and the photo gain can be written as

$$J_{\text{dark}} = (1 + \beta) J_{CBO} \approx \beta J_s \quad (R8)$$

$$G = \eta \cdot \beta \quad (R9)$$

where η is the external quantum efficiency of the base-collector junction.

As for $V_{\text{bias}} < 0$ V, e. g. $V_{\text{bias}} = -1.5$ V, $V_{\text{bi-CB}}$, $x_{\text{n-CB}}$, $L_{\text{eff-B}}$, $L_{\text{eff-E}}$ and ΔE_g can be calculated according to the parameters in Table R-1

$$\begin{aligned} V_{\text{bi-CB}} &= \frac{k_B T}{e} \ln \left(\frac{N_C N_B}{n_i^2} \right) = 0.24 \text{ V} \\ x_{\text{n-CB}} &= \left[\frac{2\epsilon_s (V_{\text{bi-CB}} + V_{\text{CB}}) N_C}{e} \frac{1}{N_B N_C + N_B} \right]^{\frac{1}{2}} = 138 \text{ nm} \\ L_{\text{eff-B}} &= x_B = x_{B0} - x_{\text{n-CB}} = 22 \text{ nm} \\ L_{\text{eff-E}} &= L_E = 592000 \text{ nm} \\ \Delta E_g &= 0.34 \text{ eV} \end{aligned}$$

Considering an external quantum efficiency (η) of 10%-30% in the base-collector junction [Ref. R10, R11], and then β , J_{dark} and G can be obtained as

$$\begin{aligned} \beta &= \frac{N_E D_B L_{\text{eff-E}} e^{\frac{\Delta E_g}{k_B T}}}{N_B D_E L_{\text{eff-B}}} = 2550 \\ J_{\text{dark}} &= 0.19 \text{ mA} \cdot \text{cm}^{-2} \\ G &\approx 300 \sim 800 \end{aligned}$$

The theoretical calculated value of the gain is completely compatible with the measurements reported in the manuscript.

Table R-1 Parameters for gain calculation

Parameters	Values	Referecnes
D_{Si}	35 cm ² /s	Ref. R9
$D_{PbS-TBAI}$	1.32e-4 cm ² /s	Ref. R12
$D_{PbS-EDT}$	1.81e-8 cm ² /s	Ref. R13
L_{Si}	592000 nm	Ref. R9, R14
$L_{PbS-TBAI}$	290 nm	Ref. R12

$L_{PbS-EDT}$	4 nm	Ref. R13
n_{i-CQD}	$1e14 \text{ cm}^{-3}$	Ref. R15

- Ref. R9 Neamen, Donald A. *Semiconductor physics and devices*. McGraw-Hill, (2012).
- Ref. R10 K. M. Xu, et al. *Inverted Si:PbS colloidal quantum dot heterojunction-based infrared photodetector*. *ACS Appl. Mater. Inter.*, 12, 15414-15421 (2020).
- Ref. R11 X. Xiao, et al. *High quality silicon: colloidal quantum dot heterojunction based infrared photodetector*. *Appl. Phys. Lett.*, 116, 101102 (2020).
- Ref. R12 L. L. Hu, et al. *Temperature- and ligand-dependent carrier transport dynamics in photovoltaic PbS colloidal quantum dot thin films using diffusion-wave methods*. *Sol. Energ. Mat. Sol. C.*, 164, 135-145 (2017).
- Ref. R13 M. J. Speirs, et al. *Temperature dependent behaviour of lead sulfide quantum dot solar cells and films*. *Energ. Environ. Sci.*, 9, 2916-2924 (2016).
- Ref. R14 V. Adinolfi, & E. H. Sargent, *Photovoltage field-effect transistors*, *Nature*, 542, 324-327 (2017).
- Ref. R15 O. Voznyy, et al. *A charge-orbital balance picture of doping in colloidal quantum dot solids*. *ACS Nano*, 6, 8448-8455 (2012).

2.2 For $V_{ce} > 0$ the sentence “the photoelectrons will be blocked off at the n-CQD:p-CQD junction and a negative photovoltage is produced at the interface, shrinks the depletion region of the n-CQD:p-CQD junction and extract holes from p-CQD to the base region (n-CQD)” is not clear. According to the band diagram of Fig. 3b, it is not clear why only the photoelectrons should be blocked at the n-CQD:p-CQD interface. Moreover, the authors assume that holes are transferred from the n-CQD to the p-Si by tunneling, but this assumption is not substantiated by experimental data nor theoretical analysis (i.e. give the low barrier height at the Si:CQD heterojunction also thermal emission phenomena should be considered). Finally, it is not clear why, at $V_{ce} > 0$, the authors still observe a photocurrent gain > 1 .

Response: We do thank the reviewer for his/her suggestion and we do understand that the reviewer wonders about the high photocurrent gain at $V_{bias} > 0$ V. We will firstly explain the blocking and transferring processes of the photo-induced carriers at the two p-n junctions and then theoretically calculate the gain at $V_{bias} > 0$ V to show the reason.

i. Blocking and transferring processes of the photo-induced carriers at the two p-n junctions

For $V_{bias} > 0$ V, p-CQD is the emitter, n-CQD is the base and p-Si is the collector. The p-Si:n-CQD heterojunction is reverse biased while the n-CQD:p-CQD junction is forward biased. Photoelectrons should overcome the built-in potential barrier to transfer from n-CQD to p-CQD, so it is likely that photoelectrons can be blocked at the n-CQD:p-CQD interface at a low V_{bias} . As for photoholes, of cause they can be blocked at the n-CQD:p-Si interface due to the valence band offset induced barrier. However, the electric field at the n-CQD:p-Si interface is enhanced by the reverse biased heterojunction, so photoholes are possible to overcome the barrier by tunneling or thermal emission. The purpose of using Fig. 3b is to show that a low positive V_{bias} , photoelectrons and photoholes can be blocked at the n-CQD:p-CQD and n-CQD:p-Si interfaces, respectively, but photoholes are possible to overcome the barrier due to the enhanced electric field at the

n-CQD:p-Si interface by the reverse biased heterojunction. We have modified several statements in this paragraph to make it clearer.

ii. **Theoretical calculation of the gain at $V_{bias} > 0$ V**

The gain at $V_{bias} > 0$ V can still be calculated according to eqs. R2-R9. It is worthy to mention that as for bulk materials, the $\frac{D_B L_{eff-E}}{D_E L_{eff-B}}$ term in eq. R7 is negligible and eq. R7 is simplified to the following expression:

$$\beta = \frac{N_E}{N_B} e^{\frac{\Delta E_g}{k_B T}} \quad (R10)$$

If using eq. R10 to calculate the gain, the obtained value will not exceed 1. This may be the reason why the reviewer thinks the gain at $V_{bias} > 0$ should not exceed 1. However, as for the zero dimensional materials (e. g. QDs), the diffusion coefficient and the diffusion length are significantly different from the bulk materials, and they cannot be reduced for the calculation of β . According to the parameters in Table R-1, as for $V_{bias} > 0$ V, e. g. $V_{bias} = 1.5$ V, V_{bi-CB} , x_{n-CB} , L_{eff-B} , L_{eff-E} and ΔE_g can be calculated as follows

$$V_{bi-CB} = 0.11 \text{ V}$$

$$x_{n-CB} = \left[\frac{2\epsilon_C \epsilon_B (V_{bi-CB} + V_{CB}) N_C}{e N_B \epsilon_C N_C + \epsilon_B N_B} \right]^{\frac{1}{2}} = 57 \text{ nm}$$

$$L_{eff-B} = x_B = x_{B0} - x_{n-CB} = 103 \text{ nm}$$

$$L_{eff-E} = L_E = 4 \text{ nm}$$

$$\Delta E_g = 0 \text{ eV}$$

Considering an external quantum efficiency (η) of 30%-80% in the base-collector junction [Ref. R16], and then β , J_{dark} and G can be obtained as

$$\beta = \frac{N_E D_B L_{eff-E}}{N_B D_E L_{eff-B}} = 283$$

$$J_{dark} = 0.02 \text{ mA} \cdot \text{cm}^{-2}$$

$$G \approx 100 \sim 200$$

The theoretical calculated value of the gain is completely compatible with the measurements reported in the manuscript.

Ref. R16 K. Lu, et al. Efficient PbS quantum dot solar cells employing a conventional structure, J. Mater. Chem. A, 5, 23960-23966 (2017).

2.3 *The device current shows a non-monotonic behavior for $V_{ce} > 0$ both in dark and under illumination. The authors discuss this behavior, but their explanation seems speculative. No numerical, experimental, or theoretical analysis are provided to substantiate the authors' assumptions. Also, references are lacking.*

Response: We thank the reviewer for the constructive suggestions. We have added more analysis and references to further support our explanation in the manuscript.

The non-monotonic behavior of the device current detected at $V_{bias} > 0$ V is due to the large injection effect. The device current density can be expressed by [Ref. R9]

$$J = (1 + \beta) J_L \approx e G_L W \beta \quad (R11)$$

where J_L is the photocurrent density generating in the base-collector junction, G_L is the generation rate of excess carriers under illumination, and W is the space charge region width of

the base-collector junction.

At the case of low injection, both W and β will increase with V_{bias} increasing. When V_{bias} exceeds a threshold, the injected minority carrier concentration will rapidly increase, which can be larger than the majority carrier concentration and lead to more sharp increase of J_{nE} than J_{pE} . According to eq. R3, β will decrease at the case of large injection with V_{bias} increasing. This is one reason why J shows a non-monotonic behavior for $V_{bias} > 0$ V. The second reason is that at the case of high injection, the minority carrier in the space charge region of the base cannot be completely depleted, which increases the neutral region width in the base and moves the depletion region edge in the base towards the collector [Ref. R17, R18]. This effect will also reduce β , and lead to the non-monotonic behavior of the device current. In addition, increasing V_{bias} can enhance the electronic field of the Si:CQD heterojunction, and the photoinduced electrons are easier to be extracted. The photoinduced electrons are blocked off at the interface of the n-CQD:p-CQD junction to increase photovoltage initially. However, with V_{bias} increasing, the barrier of the n-CQD:p-CQD junction will be gradually lowered down. More photoelectrons can tunnel across the barrier of the n-CQD:p-CQD junction and reduce the photovoltage, which also results in β decrease.

It is worthy to mention that at $V_{bias} > 0$ V, p-Si acts as the collector and at $V_{bias} < 0$ V, p-CQD acts as the collector. The doping concentration of p-Si ($5 \times 10^{15} \text{ cm}^{-3}$) is far below the doping concentration of p-CQD ($1 \times 10^{17} \text{ cm}^{-3}$) and the thickness of p-Si (520 μm) is much more than the thickness of p-CQD (~40 nm). Therefore, the large injection effect is prominent for $V_{bias} > 0$ V and at the range of $-1.5 \text{ V} \leq V_{bias} \leq 1.5 \text{ V}$, the non-monotonic behavior of the device current is only detected at the positive voltages.

Ref. R17 K. Tabatabaie-Alavi, et al. (In,Ga)As/(In,Al)As heterojunction lateral PNP transistors, International Electron Devices Meeting (1982).

Ref. R18 D. L. Harnome, et al. SiGe-base PNP transistors fabricated with n-type UHV/CVD LTE in a "No Dt" process, Symposium on VLSI Technology (1990).

2.4 Fig. 4b and the relative discussion are misleading, since it seems that visible photons are absorbed only at the p-Si:n-CQD interface. The sentence "At a positive bias, photo-induced electron-hole pairs will generate at both sides of the reversed heterojunction" is also misleading, since it neglects carrier generation occurring in the forward bias homojunction.

Response: We are sorry that the reviewer misunderstands the key point of this paragraph. At a positive bias, of cause photo-induced electron-hole pairs will also generate at the p-CQD:n-CQD homojunction. However, the homojunction is forward biased and the space charge region width (W) narrows with V_{bias} increasing as follows

$$W = \left[\frac{2\varepsilon_s}{q} \cdot \frac{N_A + N_D}{N_A N_D} (V_{bi} - V_{bias}) \right]^{\frac{1}{2}} \quad (\text{R12})$$

where ε_s is the permittivity of CQD, N_A and N_D are the acceptor and donor concentrations of p-CQD and n-CQD, respectively, and V_{bi} is the built-in potential. The photo-induced electron-hole pairs will recombine soon at this case, and the photocurrent generated at the forward homojunction is negligible compared with the one at the reversed heterojunction. Consequently, we do not consider the carrier generation and recombination occurring in the forward bias homojunction.

In order to prevent misunderstanding, we have modified the statements to make the manuscript more precise.

3. *The authors claim to have reached very high detectivity values (4.73×10^{13} Jones in the SWIR and 2.47×10^{13} Jones in the visible), but it is unclear how they evaluated the noise current. First, it seems the authors assume shot noise is the only type of noise affecting the device performance while, in the “Temporal performance and self-tunable spectral response” paragraph, the authors discuss the need to also consider other types of noise in their calculation. In this second case it is not clear how the authors could obtain a detectivity as high as 1.33×10^{13} Jones. In fact, when dealing with flicker noise, it is important to consider the integral of the noise current over the whole bandwidth of the device. Looking at the normalization noise current reported in the supplementary information it is clear that the device is strongly affected by flicker noise, but the authors apparently consider only the noise current measured at a single frequency corresponding to the cut-off frequency of the device. In this case, the authors are assuming to couple the device to an ideal bandpass filter, centered at the cut-off frequency. This assumption seems unrealistic and should have been explained in more detail in the manuscript. Moreover, the reported noise data has been measured at very low V_{ce} ($-0.1V$) while the responsivity is always discussed at higher voltages. Since the flicker noise current may be nonlinearly dependent on the DC current flowing in the device, the noise should be evaluated at the same bias at which the responsivity is measured. My opinion is that the calculated detectivity is way too optimistic and a thorough noise analysis should be conducted before claiming such high detectivity values.*

Response: We do thank the reviewer’s professional comments and suggestions and we are very sorry for the infelicitous phrasing which leads to misunderstanding. We have revised the statement in the “Temporal performance and self-tunable spectral response” paragraph.

As the reviewer stated, we did assume the read circuit performance. It is attributed to the high compatibility with Si technology of quantum dots. The detailed explanation of the assumption is as follows.

The specific detectivity of the device is

$$D^* = \frac{\sqrt{A\Delta f}}{NEP} = \frac{R\sqrt{A\Delta f}}{I_{ntotal}} = \frac{R\sqrt{A\Delta f}}{\sqrt{2q(GNDCD \cdot A + I_{\phi})M^2F \cdot \Delta f + I_{nROIC}^2}} \quad (R13)$$

where I_{ntotal} is the noise current, $GNDCD = \frac{I_{dark}}{M \cdot A}$, M is the gain, F is the excess noise, A is photosensitive element area, I_{ϕ} is the background luminous flux, Δf is the noise bandwidth, I_{nROIC} is the circuit noise current. Generally, the background luminous flux is relatively small, about 10^8 photons/S/cm². For our device area in the manuscript, the current is in the order of sub femto ampere, which can be ignored. Considering the high compatibility with Si technology of quantum dots and the strikingly similar noise figure of our device and the silicon-only device [Supplementary Information in Ref. R1], we assumed that the number of circuit noise electrons stayed below single figures, which can be ignored in the multiplier detector. In these cases, the specific detectivity of the device is

$$D^* = \frac{\sqrt{A\Delta f}}{NEP} = \frac{R\sqrt{A\Delta f}}{I_{ntotal}} = \frac{R\sqrt{A\Delta f}}{\sqrt{2qGNDCD \cdot A \cdot M^2 \cdot F \cdot \Delta f}} = \frac{R\sqrt{A}}{\sqrt{2qI_{dark} \cdot M \cdot F}} = \frac{R}{\sqrt{I_n^{*2}}} \quad (R14)$$

Under this condition, the device bandwidth can be ignored. In addition, it is a general way to use shot noise to evaluate the noise current and calculate the specific detectivity by using the normalization noise current (I_n^*) at the cut-off frequency (f_T). Lots of references can support this method such as Ref. R1, R19-21.

We agree the reviewer's opinion that the noise should be evaluated at the same bias at which the responsivity is measured. The maximum specific detectivity extracted by the dark current method is 4.73×10^{13} Jones with V_{bias} of -0.4 V at 1,550 nm, and the corresponding responsivity is 125 $A \cdot W^{-1}$. I_n^* was measured at the same bias voltage ($V_{bias} = -0.4$ V) for the device (Fig. S7). The noise spectrum of a detector is a plot of noise current spectral density versus frequency that is used to determine the magnitude of noise at the frequency at which the photodetector operates [Ref. R22]. The cut-off frequency of the PVTRI can be calculated by the equation: $f_T = 1/2\pi\tau$, where τ is the response time. For $V_{bias} < 0$ V, the response time is about 50 μs (Fig. S6a) and the corresponding normalization noise current is 1.17×10^{-11} $AHz^{-0.5}cm^{-0.5}$. The extracted specific detectivity of the PVTRI is up to 1.07×10^{13} Jones, which is of the same magnitude as the value calculated by the dark current method.

Ref. R19 S. Ghosh, et al. Enhanced responsivity and detectivity of fast WSe₂ phototransistor using electrostatically tunable in-plane lateral p-n homojunction, Nat. Commun., 12, 3336 (2021).

Ref. R20 A. De Iacovo, et al. Noise performance of PbS colloidal quantum dot photodetectors, Appl. Phys. Lett., 111, 211104 (2017).

Ref. R21 X. Gong, et al. High-detectivity polymer photodetectors with spectral response from 300 nm to 1450 nm, Science, 325, 1665-1667 (2009).

Ref. R22 R. Saran and R. J. Curry. Lead sulphide nanocrystal photodetector technologies, Nat. Photonics, 10, 81-92 (2016).

4. The authors claim that "The distinguishable response time between SWIR and visible light excitation endows our PVTRI the self-tunable capability to distinguish SWIR and visible signals in one chip, which is promising in the cutting-edge optoelectronic applications". This assumption should be discussed in more details, since it is not clear how the authors would employ the proposed device in a real-life application in order to distinguish between different wavelengths. What happens if SWIR and visible light reach the device at the same time?

Response: We thank the reviewer for this suggestion. We have added more discussion in the revised manuscript.

- i. If SWIR and visible light do not reach the device at the same time, they can be distinguished by the falling edge tail of the $I_{ph}-t$ plot at the case of $V_{bias} > 0$ V. There is distinguishable response time under SWIR and visible light illumination for the proposed device. At $V_{bias} > 0$ V, the response time (falling edge) of visible light is significantly faster than the one of SWIR.
- ii. If SWIR and visible light reach the device at the same time, they can be distinguished by double measurements on both sides of the device. To begin with, the position of the device can be adjusted to make the incident light illuminate on the Si substrate. At this case, the Al electrode should not cover the whole substrate for a real-life application. Since the Si substrate is hundreds of microns, it can act as a visible light filter and we can get the signal of SWIR by this measurement. Then, we can reverse the device and make the incident light

illuminate on the CQD. The mixed signal of SWIR and visible light can be got by this second measurement. At last, we can get the signal of visible light by curve extraction from the above two plots.

5. *Some references are out of context and should be removed.*

Response: We do thank the reviewer for his/her earnestness and rigor. We have carefully checked all the references and removed some of them which seem to be out of context.

6. *The paper is poorly written and should be thoroughly checked for language inconsistencies.*

Response: We have thoroughly checked the manuscript, modified the unclear statements and corrected the confusing language. We hope the revised version can meet with the reviewer's approval.

Reviewers' Comments:

Reviewer #1:

Remarks to the Author:

The authors have addressed all my comments.

Reviewer #2:

Remarks to the Author:

Dear authors,

I really appreciate the work you have done to improve the paper. The several additions and modifications you made, enhanced both the paper clarity and its scientific rigor, allowing it to reach the quality standards required by Nature Communications.

I agree with many of the arguments you brought up in your rebuttal but I still think that the detectivity calculation should have been explained in more detail in the paper. I am aware that many authors account only for the shot noise when calculating the detectivity, but I still think that, in this particular case, this approach leads to an overestimation of the D^* , since the proposed device is definitely dominated by flicker noise.

I strongly suggest the authors add at least a short discussion, justifying their calculation approach as they did in the rebuttal letter.

For this reason, I recommend the paper for publication after a minor revision.

Response to reviewers:

We would like to thank the reviewers for their highly constructive reviews of our manuscript. In the following, we address each of the reviewers' comments in detail.

Point-to-Point Responses

Reviewer #1 (Remarks to the Author):

The authors have addressed all my comments.

Response: We do thank the reviewer for his/her recognition to our work.

Reviewer #2 (Remarks to the Author):

Dear authors, I really appreciate the work you have done to improve the paper. The several additions and modifications you made, enhanced both the paper clarity and its scientific rigor, allowing it to reach the quality standards required by Nature Communications. I agree with many of the arguments you brought up in your rebuttal but I still think that the detectivity calculation should have been explained in more detail in the paper. I am aware that many authors account only for the shot noise when calculating the detectivity, but I still think that, in this particular case, this approach leads to an overestimation of the D^ , since the proposed device is definitely dominated by flicker noise. I strongly suggest the authors add at least a short discussion, justifying their calculation approach as they did in the rebuttal letter. For this reason, I recommend the paper for publication after a minor revision.*

Response: We do thank the reviewer for his/her recognition to our work. The calculated specific detectivity may be overestimated if only accounting for the shot noise even though this method is commonly used [Ref. R1-4]. In this manuscript, the specific detectivity is calculated by using the normalization noise current (I_n^*) at the cut-off frequency (f_T). In order to justify this calculation approach, we have separately extracted the normalization shot noise (I_{ns}^*), the generation-recombination (G-R) noise (I_{ngr}^*) and the 1/f noise (flicker noise, I_{nf}^*) according to eqs. R1-R3, calculated the total noise current (I_{ntotal}^*) and compared it with the measured one.

$$I_{ns}^* = \sqrt{2qI_d} \quad (R1)$$

$$I_{ngr}^* = \sqrt{\frac{4qMI_d}{1 + (2\pi f\tau)^2}} \quad (R2)$$

$$I_{nf}^* = \sqrt{\frac{cI_d^\alpha}{f^\beta}} \quad (R3)$$

$$I_{ntotal}^* = \sqrt{(I_{ns}^*)^2 + (I_{ngr}^*)^2 + (I_{nf}^*)^2} \quad (R4)$$

where I_d is the dark current at $V_{bias} = -0.4$ V, M is the photocurrent gain at $V_{bias} = -0.4$ V under 1550 nm illumination, f is the work frequency of the device, τ is the average carrier lifetime, C , α and β are constant. α and β are typically set at 2 and 1, respectively [Ref. R5] and c is chosen as 3×10^{-8} according to the measured normalization noise current.

As shown in Fig. R-1, the calculated total normalization noise current (red curve) is basic anastomotic with the measured one (black dots). In addition, the calculated total normalization

noise current at the cut-off frequency (f_T) is $1.01 \times 10^{-11} \text{ AHz}^{-0.5}\text{cm}^{-1}$, which is very close to the measured one ($1.17 \times 10^{-11} \text{ AHz}^{-0.5}\text{cm}^{-1}$). As a result, the calculation approach in this manuscript is accurate for D^* estimation.

Fig. R-1 Normalization noise current (I_n^*) was measured as a function of frequency at bias voltage $V_{bias} = -0.4 \text{ V}$ for the PVTRI. The measured noise, calculated 1/f noise, shot noise, and G-R noise limit are also included for reference.

Ref. R1 Adinolfi, V. and Sargent, E. H. Photovoltage field-effect transistors. *Nature* 542, 324-327 (2017).

Ref. R2 S. Ghosh, et al. Enhanced responsivity and detectivity of fast WSe_2 phototransistor using electrostatically tunable in-plane lateral p-n homojunction, *Nat. Commun.*, 12, 3336 (2021).

Ref. R3 A. De Iacovo, et al. Noise performance of PbS colloidal quantum dot photodetectors, *Appl. Phys. Lett.*, 111, 211104 (2017).

Ref. R4 X. Gong, et al. High-detectivity polymer photodetectors with spectral response from 300 nm to 1450 nm, *Science*, 325, 1665-1667 (2009).

Ref. R5 C. Chen, et al. One-dimensional Sb_2Se_3 enabling a highly flexible photodiode for light-source-free heart rate detection, *ACS Photonics*, 7, 352-360 (2020).